# Google Earth Geoscience Video Library (GEGVL): Organizing Geoscience Videos in a Google Earth Environment to Support Fieldwork Teaching Methodology in Earth Science

Ning Wang *, Robert J. Stern, Mary L. Urquhart and Katherine M. Seals

School of Natural Sciences and Mathematics, University of Texas at Dallas, Richardson, TX 75080, USA; rjstern@utdallas.edu (R.J.S.); urquhart@utdallas.edu (M.L.U.); katie.seals@utdallas.edu (K.M.S.)
* Correspondence: wangningapply@gmail.com; Tel.:+1-469-258-9490

**Abstract:** Fieldwork teaching methodology (FTM) and active learning are effective strategies for geoscience education. However, traditional field trips require significant resources, time, physical abilities, and the expertise of teachers. In this study, we provide a supplementary virtual field trip experience by showing how different kinds of geoscience videos can be spatially organized into one digital interactive virtual environment. Here, we present the Google Earth Geoscience Video Library (GEGVL) which uses Google Earth and location-specific videos about Earth events, to create a virtual field-based learning experience. Using Google Earth, GEGVL organizes field-based videos by location and links pertinent non-field-based videos, and allows users to roam the globe in search of geoscientific videos that are pertinent to them or their students. Currently, GEGVL contains 150 videos organized into ten different geoscience disciplines: Plate Tectonics, Minerals, Structural Geology, Metamorphism, Magmatism, Hydrology, Environmental Science, Sedimentology, Paleontology, and Paleomagnetism. Despite stability challenges with Google Earth integration, results of user surveys among lower-division undergraduates show that the design logic of GEGVL is a promising virtual field-based learning organizer for increasing students' interest in and helping them learn about Earth sciences.

**Keywords:** multimedia; geoscience videos; geoscience education; GEGVL; Earth systems science education; educational technology; field-based learning; active learning

## 1. Introduction

Improving student understanding of Earth systems, the environment, and how people interact with these is crucial to better respond to many of the challenges that we face today [1–3]. Such understanding is also important for developing geoethics and achieving sustainability [4–7], but it is hard to foster the understanding of Earth systems and the complexity of geological processes [8–10]. To achieve these goals, we must find new ways to increase student interest in Earth science, foster the development of a holistic view of Earth systems, and increase students' knowledge of geoscience topics [9,11]. However, Earth science education in many schools and universities falls short in terms of these efforts, and many teachers are not well-trained to teach Earth science [12]. Orion [5], Lombardi [13], and Vasconcelos [4] identified the important roles that active learning and outdoor education play in these efforts because these methods especially engage students and enhance learning outcomes [3,14,15]. Nevertheless, these strategies have drawbacks, especially in terms of implementation. Field trips can be difficult due to high costs, limited expertise of teachers, location of the school, and physical abilities of teachers and students [16,17]. Many active learning strategies such as inquiry-based or project-based learning can be also time-consuming to design and manage, sometimes with high cost or needing special equipment or training [18–22]. As a complementary tool, videos could be effective in increasing students' interest and knowledge [23–27]. The multimedia nature

of videos is especially powerful for teaching and learning about the Earth because these combine spoken words (and closed-captioned text) and dynamic visual information can engage viewers and at the same time could use multimedia designs to manage cognitive load [28]. Moreover, although there are many other digital solutions such as virtual field trips (360 camera-based [29], VR-based [17,30], simulation-based [31], or data-visualization-based [32] methods, videos are still the most accessible and easy-to-implement solution for teachers.

The abundance of online videos on various geoscience topics now allows students to choose what they want to learn (so-called mastery learning) and support their active learning. In addition, video is a very useful and accessible multimedia tool for geoscience learning, as these provide visualizations of complex geologic concepts, give virtual experiences, and are often freely available on YouTube [33]. Well-designed and targeted videos can also help manage cognitive load and potentially increase the engagement of students [34,35]. However, there are many geoscientific videos of varying scientific quality, so there is a real challenge to filter and organize these so that instructors and students can find good ones easily [36]. In this paper, we show how we organized geoscience videos using the well-known and accessible Google Earth platform. The resulting Google Earth Geoscience Video Library (GEGVL) is designed to integrate geoscience videos in a virtual Earth environment in order to engage students and help instructors quickly find suitable, scientifically robust videos to help them teach Earth science.

The design of the GEGVL organizes a variety of field-based geoscience videos into a virtual Earth environment and allows users to freely explore "Earth stories" that are specific to a place. With the development of inexpensive video-making technology and smartphones, and the rise of popular video portals such as YouTube, the amount and diversity of geoscience videos is increasing rapidly. Geoscience educational videos now cover a great variety of geoscience topics in space and time from multiple perspectives, making these a rich resource for teachers and learners. A considerable challenge is how to assure scientific quality and organize these in an easily accessible geospatial library. We surveyed 71 undergraduate geoscience students about their experience using GEGVL and their opinions about its design. The results imply that this is a promising way to engage students to learn about the Earth. Despite its promise, GEGVL challenges remain for use in the classroom due to incompatibilities with the Google Earth platform, which causes GEGVL to crash after clicking on a few videos. Nevertheless, the GEGVL experiment is the first attempt of which we are aware to provide a general structure to organize videos effectively to serve Earth science education.

In summary, the advantages of GEGVL include the following features:

- Creating an interactive, informative, and exploratory environment to trigger students' interest in exploring Earth science knowledge;
- Organizing and screening science-accurate geoscience videos to support easier and faster geoscience video searching.

In this paper, we first explain the GEGVL design and then present the results of classroom assessment before discussing some implications of these efforts. Improved designs incorporating "lessons learned" from our GEGVL experience could result in the development of powerful educational tools.

## 2. GEGVL Design

GEGVL was designed to serve K-12 teachers and college instructors by providing a more efficient and vetted geoscience video search method. GEGVL is expected to serve teachers as a library to provide the platform and information to teachers to easily navigate and find relevant and reliable videos to create suitable video-based teaching sections or assignments for students more efficiently. The diversity and abundance of geoscientific educational (GeoEd) videos present teachers and students with more choices but also more challenges. Current research shows that the two most mentioned concerns of K-12 teachers are video search time and the trustworthiness of content [36,37]. Many good GeoEd

videos are too recent or not popular enough to be shown in the first several pages of search results [33]. Moreover, many GeoEd videos are not scientifically accurate or are otherwise inappropriate [37]. Searching for pertinent videos on Google or YouTube can be very time-consuming [38]. For students and some teachers who do not yet have a good understanding of geosciences, their YouTube or Google search for geoscience topics can find misinformation-filled videos (e.g., creationist, flat Earth, climate change deniers) and can have their learning impeded by watching these [39]. Search requires knowledge of the search space, especially for videos because, in contrast with scientific papers on Google Scholar, these mostly are not peer-reviewed. Therefore, the design of GEGVL includes scientific reviews by one or more geoscience experts. We performed this on an ad hoc basis, but it is clear that more thought needs to be given to how best peer-reviewed geoscience videos before putting these into GEGVL or any other educational video library so that teachers and students can be confident in video content.

Overall, the creation of GEGVL includes four major parts: (1) identifying geoscientific videos with useful, accurate content; (2) classifying these into field-based (FB) and non-field-based (non-FB) videos; (3) locating FB videos using Google Earth; and (4) providing links to pertinent non-FB videos as part of FB video links. GEGVL classifies videos into FB videos and non-FB videos. FB video contents pertain to a specific location or region that can be shown on Google Earth, e.g., Geo News videos [40], while the contents of non-FB videos are more abstract, not pertaining to any particular region (e.g., what is plate tectonics theory?). By locating FB videos on the virtual Earth and linking pertinent non-FB videos to these, we try to better engage and steer students to learn more about Earth science and scaffold their learning. This structure has the potential to organize all geoscience videos because all of these are either field-based or non-field-based. In this way, we provide users a virtual field trip experience that allows them freely explore Earth science-related concrete objects and events and connect these to more abstract concepts and processes. It is straightforward for users to start exploring all kinds of geoscience videos based on their personal interests in places, topics, or events. Thus, GEGVL is also rooted in virtual outdoor education and active learning strategies.

The FB videos in GEGVL are assigned to one of ten different geoscience disciplines: Plate Tectonics, Minerals, Structural Geology, Metamorphism, Magmatism, Hydrology, Environmental Science, Sedimentology, Paleontology, and Paleomagnetism. The fundamental components of GEGVL are field-based videos that focus on observable objects or experientable events. Many of these are familiar and relevant to people, with or without Earth science education (e.g., minerals or an earthquake). Videos present FB events and objects virtually and build on this interest to allow geoscientists to explain more abstract science concepts related to these to the engaged audience. We highlighted FB videos because people learn better when context is provided [1,16,41,42]. GEGVL builds on this to create a natural learning experience to motivate people to actively learn more at their pace in their own ways [43]. That is why GEGVL also provides non-FB videos to support their learning. Many geoscientific educational (GeoEd) videos are not about field-based events or objects but present abstract geoscientific concepts. Based on how the Earth system works, these non-FB videos are linked with the FB videos, in an effort to support students' understanding. The basic logic is to (1) start with what people know, are interested in, and are familiar with, to engage them to want to learn more geoscience; (2) give teachers and students an opportunity to explore various Earth events over various scales of time and space; and (3) allow teachers and learners to roam the Earth to freely choose what to learn. This approach can also help diverse groups with different backgrounds, interests, and geoscientific knowledge. As of April 2022, GEGVL contains 150 videos covering the ten different geoscience disciplines. We created discipline- and events-representing icons for placemarks on Google Earth, which leads to selected FB and non-FB videos.

GEGVL also considers the emotional responses and cognitive processes of students. Students have multiple ways of understanding geoscientific topics. GEGVL accomplishes this by vetting various videos made by different creators and organizing these onto one

well-known platform, Google Earth. Using multiple representatives has been proven as a very effective teaching method [44,45]. The attention, relevance, confidence, and satisfaction (ARCS) motivation model [46] emphasizes the importance of these four things for students to be motivated to learn. The virtual Earth environment allows students to explore videos about real events and objects, and this fun learning lets them feel that geoscience knowledge is relevant to them. Additionally, because there will always be new events for which new videos will be created, GEGVL can be updated to keep attracting students' interest. Such continuous triggering of temporary interests can be a key to developing students' long-term interests in geosciences [47].

GEGVL is composed of three major parts: (1) video search, review, and categorization; (2) location of placemark icons; and (3) organization of videos and related information (Elements in the Placemark Popup Page) in placemarks (Figure 1).

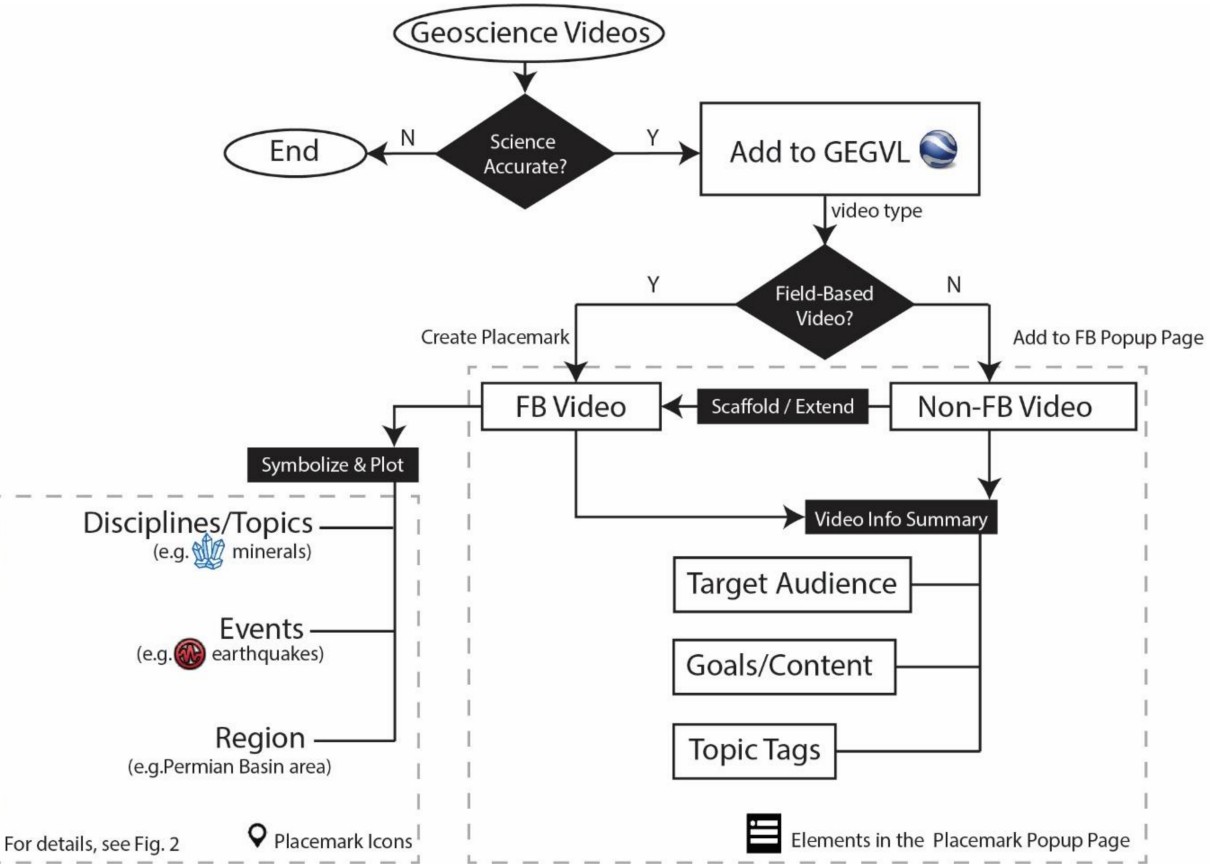

**Figure 1.** Framework of the Google Earth Geoscience Video Library.

Videos added to GEGVL are obtained in four ways. One is the videos that we make in UTD-Geoscience Studios. The second way is to directly search Earth science topics on public video platforms such as YouTube and Vimeo, etc. The third way is via recommendations of Earth science experts or through discovery. The second author, a Geoscience professor, has contributed most of the video recommendations and reviews to the current GEGVL. If some videos are beyond his expertise, we consult experts in other institutions for their opinions. The last way is to use suitable videos from Earth science individuals and groups that make GeoEd videos, such as in Earth science departments (including Christopher Spencer at Queen's University (Kingston, ON, Canada), Rob Dunbar at Stanford (Stanford, CA, USA), Matthew Malkowski at UT Austin (Austin, TX, USA), Andreas Moeller at Kansas University (Lawrence, KS, USA), Peter Clift at Louisiana State University (Baton Rouge, LA, USA), David McConnell at North Carolina State University (Raleigh, NC, USA) ), research centers (such as Science Education Resource Center at Carleton College),

and science organizations (such as IRIS, NASA and UNAVCO). In the current version of GEGVL, we are primarily searching and adding videos with location features, i.e., FB videos. As of April 2022, 103 of the 150 GeoEd videos in GEGVL were field-based. We are continuously searching, reviewing, and adding new GeoEd videos.

It is hard to quickly skim a video to decide whether or not a video matches one's needs and level; this is especially important for instructors, who are likely to be searching for material to support a particular lecture or lab topic. Thus, it is important to provide keywords and descriptions of a Geoscience video (such as targeting audience and domain knowledge topics as Elements in the Placemark Popup Page) (Figure 1). GEGVL provides at least three educational feature descriptions for each video—namely, target audience level, educational goals or content summary, and a list of Earth science topics that the video touches on. GEGVL subdivides target audiences into three ranks: beginner, intermediate, and advanced. In the GEGVL definition, beginner videos should be easily understood by audiences without any Earth science background and require very little science background to follow. The domain knowledge in beginner videos is simplified and straightforward, not going into depth and ignoring complexity. Intermediate videos are good for non-majors with a basic understanding of intro-level Earth science topics, i.e., students and others who have had an introductory course in geology. These are appropriate for advanced high school students or freshman and sophomores in community colleges or universities, and for some in the general public. Intermediate videos involve more Earth science topics and may discuss some complexity such as different possibilities and conditions, but they are still within the scope of introductory knowledge and relevant to daily life. Advanced videos are mostly made for Earth science majors, especially upper-division undergraduates and beginning graduate students. Advanced videos go into great depth and provide more details and uncertainties. As all videos in GEGVL are reviewed by at least one content expert, we list the reviewer and credit the production team in the descriptions.

Lastly, the basic functional units in GEGVL are placemarks. In the Google Earth environment, placemarks contain an icon and a content page. The icon can be plotted on the virtual Earth's surface, and it is clickable to show the content page. Each placemark is represented by a specially designed icon implying a certain geoscience topic (e.g., igneous rocks), a discipline (e.g., sedimentology), or an event (e.g., earthquake or oil drilling). The content page for each placemark contains at least one FB video that is often accompanied by one or more non-FB video(s). This straightforward design encourages users to link geological events together under an Earth system framework and allows them to explore geoscience information about certain places, topics, or events.

FB videos in GEGVL can be further categorized into event-based and object-based videos (Figure 2). Event-based FB videos are timely and mostly about natural hazards and human activities, for example, Geo News videos [40]. Natural hazards videos include earthquakes, tsunamis, and volcanic eruptions, whereas human-activity videos include scientific drilling, resources, engineering, etc. Object-based FB videos are categorized by Earth science disciplines and regions and are always about observable objects, such as mountains, aquifers, or mineral deposits. Some object-based FB videos cannot be categorized into any geoscience discipline and are about a large area since they are about regional geology. A single icon location is not appropriate in this situation, so we used polygons and different colors to represent them. By these designs, there are four ways to find any field-based video in GEGVL: (1) by exploring the interactive virtual Earth environment and clicking on placemark icons; (2) by inputting keywords in the search engine; (3) by looking through the discipline-based index of places; and (4) by using the tables function in Google Earth (Figure 3A).

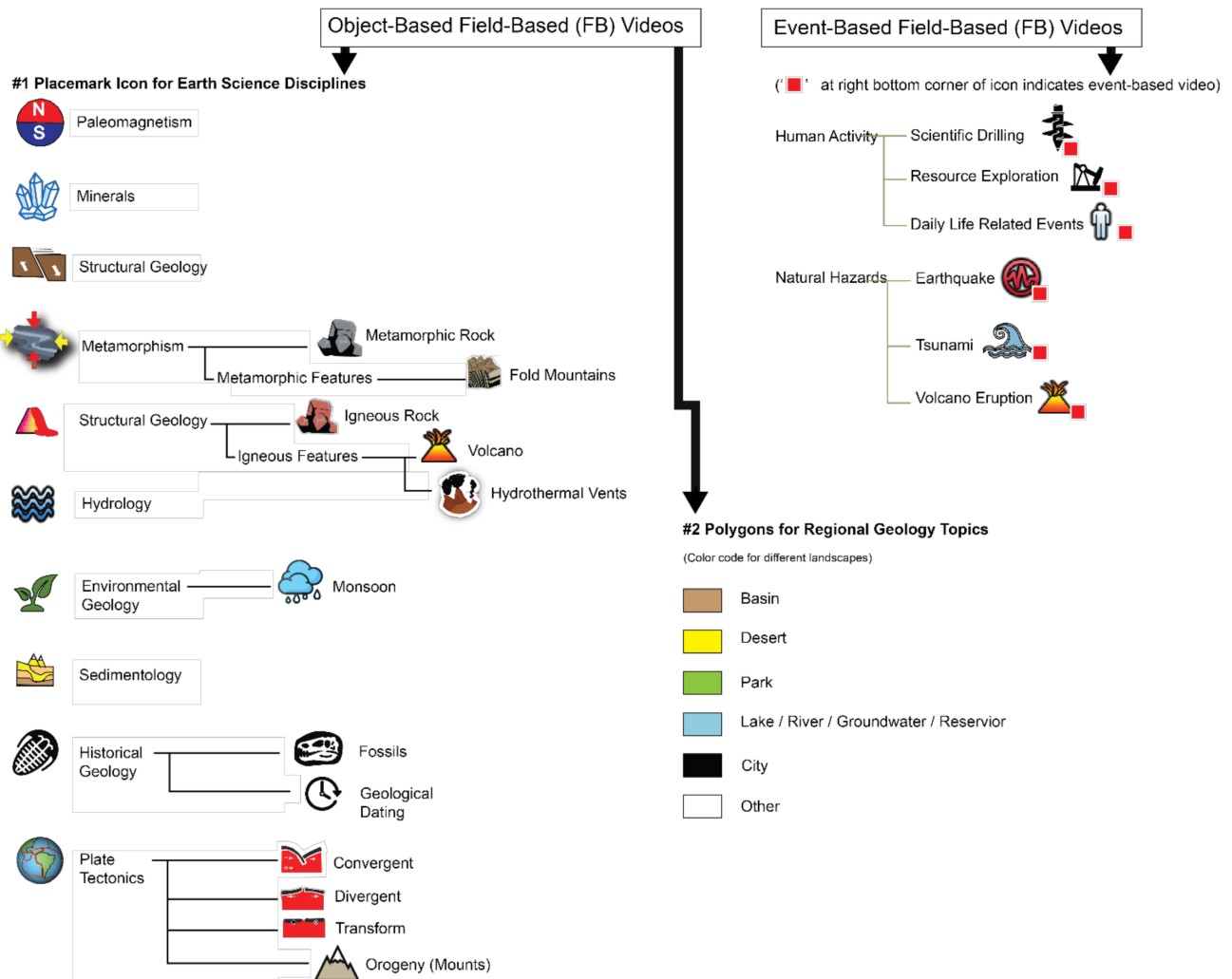

**Figure 2.** GEGVL placemark representations for two types of FB videos: object-based videos and event-based videos. Regional geology videos are represented with polygons because they cannot be categorized into any single geological discipline or represented with a single placemark.

In many cases, GEGVL adds relevant non-FB videos to explain or extend the Earth science topics presented in the FB video. Herbert [44] noted that the complexity of Earth systems is likely to cause large cognitive loads for students in their development of advanced mental models about these systems. Therefore, incorporating appropriate non-FB videos (e.g., videos explaining what is a plate) to scaffold student learning is appropriate. With the multiple representations of the same topics, multimedia designs of different videos and the abundance of non-FB videos, the design of GEGVL may help students better manage the cognitive load while learning Earth science topics. Different creators may make different videos about one topic, which provides multiple representations. Videos made by different geoscientists explaining the same topic can be very different regarding spatial scales, geological time span, angles, depth of the knowledge, details of the topic, etc. For non-FB videos, different real examples (FB videos) become the different representatives of the same concept (non-FB videos). The (Design, Functions, Tasks) model summarizes the advantages of multiple representations, and other studies show that providing multiple representations can help learners gain a deeper understanding by exposing the underlying structure of domain knowledge [33]. Although FB videos are engaging because they have more familiar contexts and provide practical examples of Earth processes, non-FB videos are important for explaining Earth Science concepts and abstract processes at different spatial scales and in deep time. GEGVL uses non-FB videos to further explain and discuss

geological concepts mentioned in the FB videos to which they are linked. For example, two non-FB videos are added to the "Hawaii Eruption 2018-05" popup page (Figure 3B). One is "Three Great Ways to Melt the Mantle" (made by UTD Geoscience Studio), and the other is "Hotspot Volcanism—A Thermal Plume" (made by IRIS Education, IRIS represents Incorporated Research Institutions for Seismology). The two non-FB videos are given to explain more about the mechanism of mantle melting and mantle plumes, which are the two important Earth science topics in the "Science Behind Hawaii Eruption 2018" FB video.

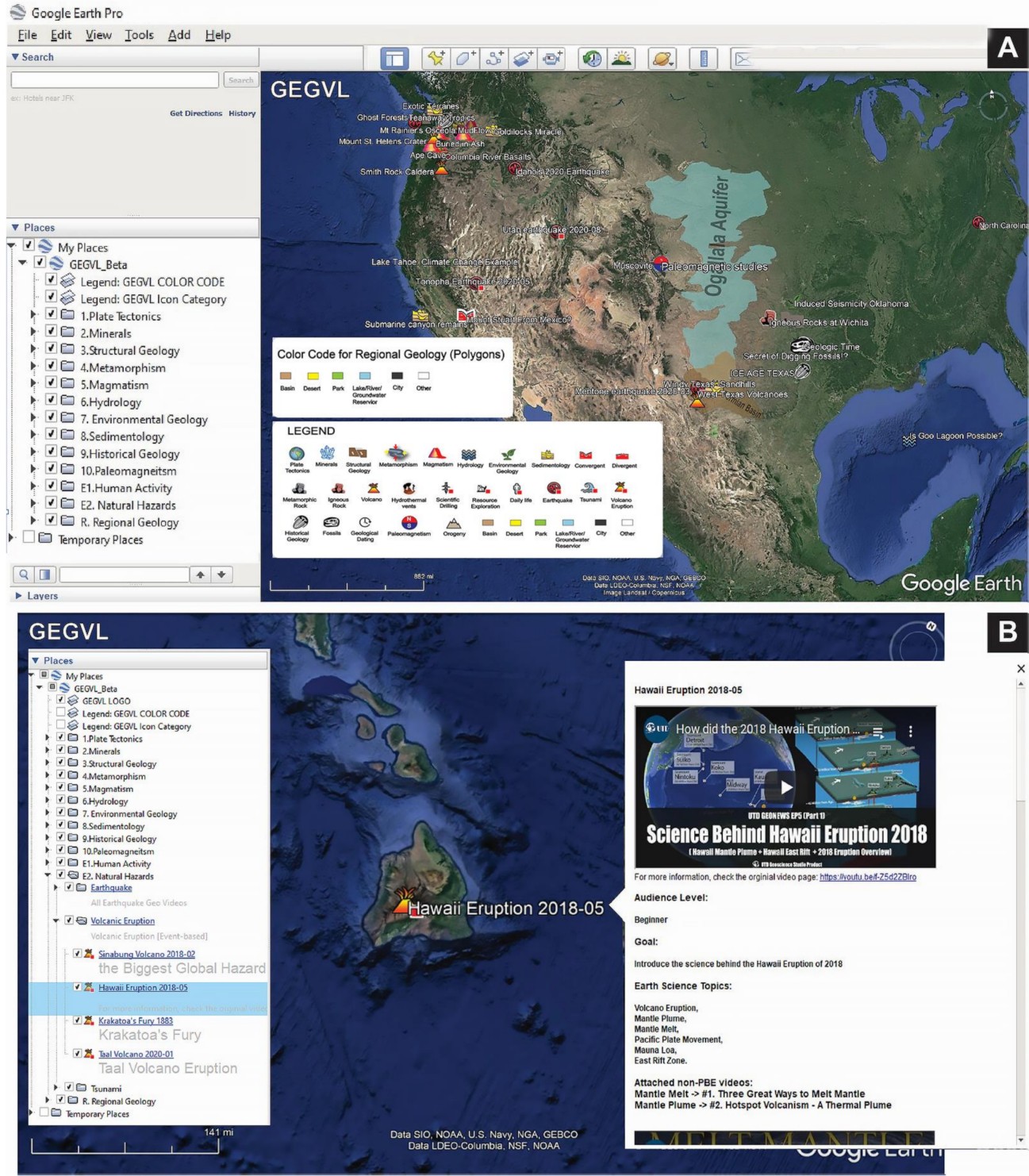

**Figure 3.** Demonstration of GEGVL panels: (**A**) GEGVL operation panel; (**B**) an example of GEGVL popup page for the "Hawaii Eruption 2018-05" placemark.

### 3. Assessment Methods

In order to determine what college students think of GEGVL, we surveyed 71 lower-division undergraduate students from the University of Texas at Dallas (Richardson, TX, USA). The students were recruited randomly from introductory geoscience classes as volunteers or signed up via emails after seeing recruiting posters on the UTD campus. All participants responded to the survey online and used their own devices to finish the survey. The online survey questionnaire was designed and sent out to the participants via the REDCap app. The questionnaire is accessible at https://redcap.link/UTDGSS-S2 (accessed on 11 May 2022). The survey includes a 4 min demonstration video of how GEGVL works (https://youtu.be/ldTS-7VUPAI, accessed on 11 May 2022) and provides the GEGVL KMZ file download link so that participants can quickly have an idea of how GEGVL operates and gain some experience using it. The survey first asked some basic demographic questions. There were 30 male and 41 female participants; of those, 7 participants reported that they often watch Earth science videos, 39 reported sometimes, and 25 reported very rare. Most (61%) of the participants were neutral (not particularly interested in Earth science), and 35% expressed that they are interested in Earth science topics, while only 3 people reported that they are not interested in Earth science at all (4%). The majors of participants include geosciences, physics, biology, art, business analysis, computer science, cognitive science, and international political economy.

### 4. Results

Responses to the experience of using GEGVL from surveyed students and teachers were, overall, positive. GEGVL was mostly liked by the surveyed students (N = 71) who want to use it in class and self-learn some Earth science topics. All students in our survey agreed that the GEGVL helped them develop a more holistic view of Earth science knowledge (63 strongly agree, 8 agree). In total, 70 students liked the GEGVL (68 strongly agreed and 2 agreed), and only 1 student did not (Figure 4). These results show that most students considered GEGVL a good tool for both classes and self-learning. However, more students believed GEGVL is better for self-learning purposes (90% versus 82% strongly agree).

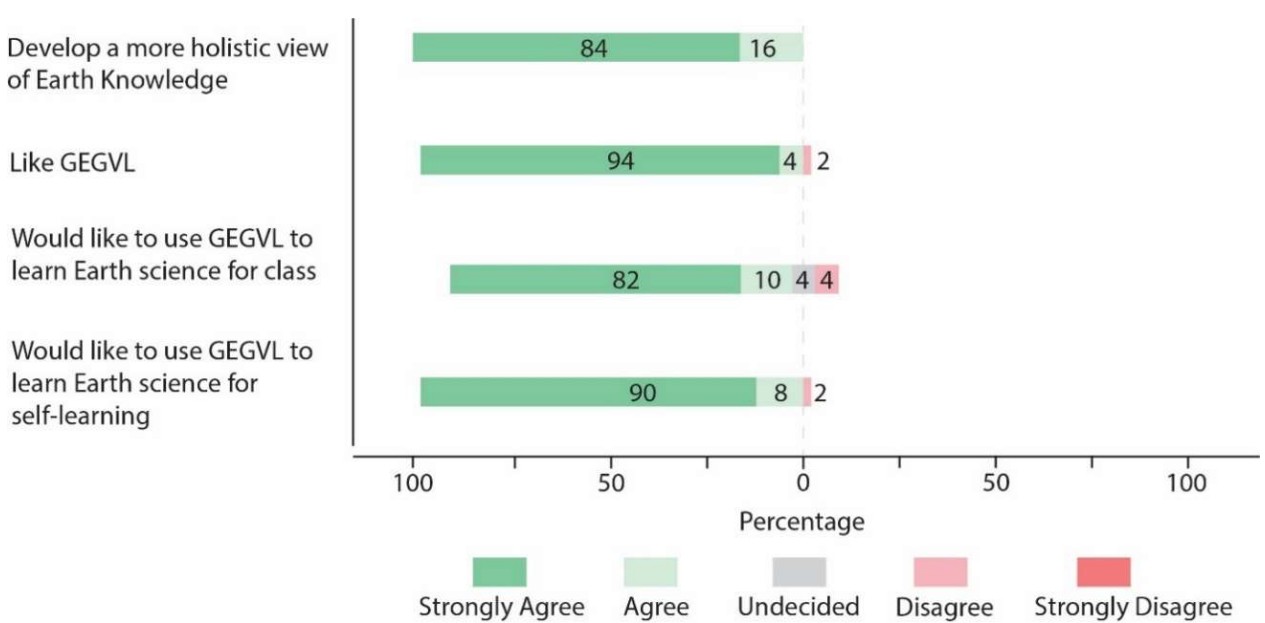

**Figure 4.** Responses to the GEGVL experience survey.

A detailed summary chart for the reasons why surveyed students like the GEGVL is shown in Figure 5. Some comments from surveyed students about the GEGVL include the following statements: "The video library does give a good ability to tie understanding of location and area to specific info."; "The benefit is the knowledge of science on the world, right from your home or location."; "It's awesome, lots of information about all places. It needs a simpler guide though."; "I like that it is interactive and informative."; "I think it is easily accessible."; "It could make for a more enjoyable learning experience immersing into new forms of learning."; "It is amazing since it has icons on the map which when clicked on moves to a video explanation of what happened in that particular place. There are also additional interesting facts, geological terms, etc., which makes learning more fun."; "This is an amazing initiative, as it makes the learning of geographical elements more realistic. It explains the concepts in a lively way." The most mentioned liked factors include (1) abundance of FB videos (mentioned 35 times), (2) exploration of geographic features (32 times), (3) easy to use (23 times), (4) big picture of Earth and feel the connections of things and events (19 times), and (5) geographic context for Earth science knowledge (18 times).

**Figure 5.** Reasons for liking GEGVL.

Students also reported some drawbacks, such as "have to download the app" (mentioned 15 times); "The details that are given as highlights on the map rather than icons can be hard to see, and an alternate method for seeing a list of all content in a browsable format would be helpful."; and "There is so much information about Earth sciences out there that watching all the possible topic videos would be difficult." There were some comments mentioned that GEGVL is unnecessary and no better than a Google search for the needed topics (Figure 6).

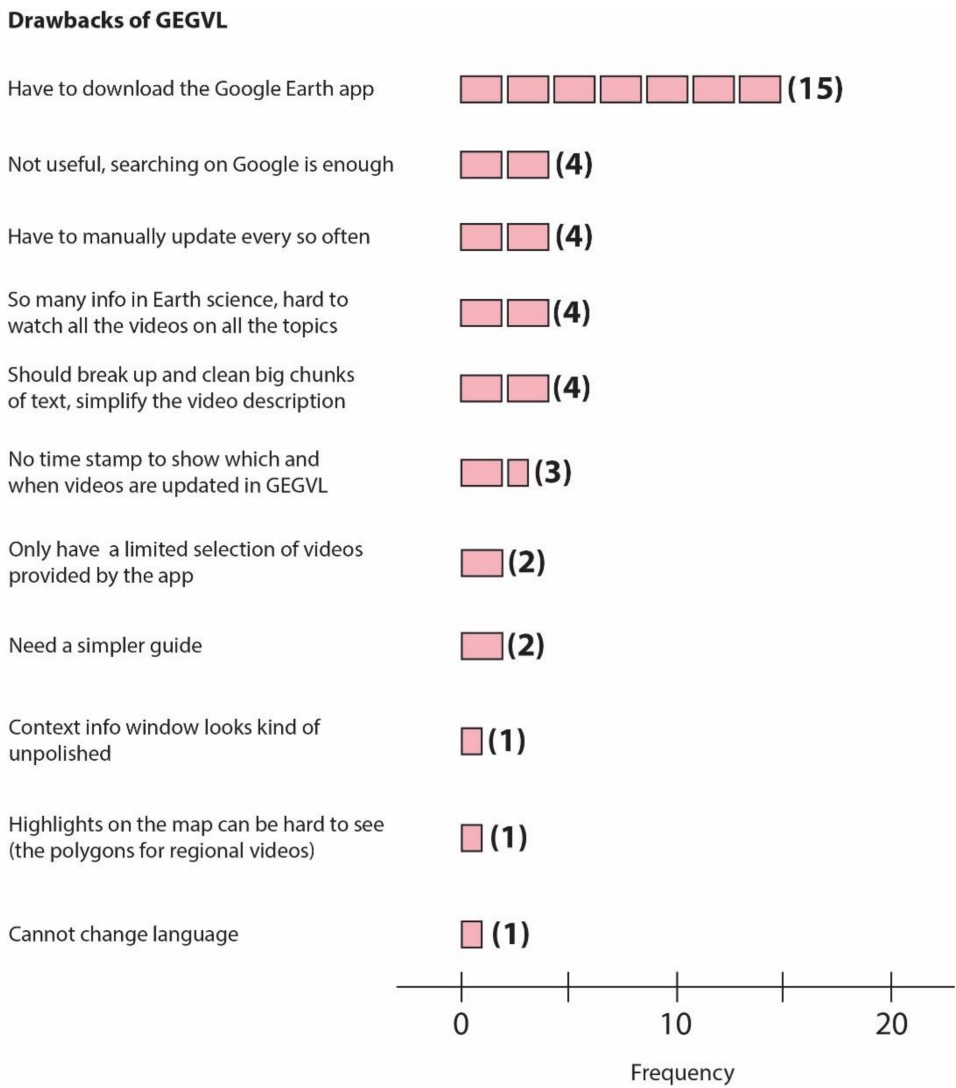

**Figure 6.** Reasons for not liking GEGVL.

## 5. Discussion

The results show that GEGVL can be effectively used to engage students in Earth science topics, with its special organization of the videos in the Google Earth environment. Herbert [44] noted that people organize their knowledge and reason about environmental issues by subconsciously constructing mental models and then manipulating them in their brains. Exploring videos on GEGVL has the potential to stimulate this activity and hypothetically provide a more intuitive way to increase their beyond-human scale experience about Earth systems and foster a better mental model for incorporating natural phenomena and integrating more information about Earth systems science. Orion and Vasconcelos [9] noted the relationships between systems thinking skills and the development of environmental insight among high school students and adults. Environmental insight requires understanding Earth subsystems such as geosphere, hydrosphere, biosphere, and atmosphere, as well as how these interact with and are affected by human activities. Field-based videos about how human activities impact the environment can support building this understanding. Following GEGVL design logic, geoscientists and educators now have a more systematic way to organize these videos to promote students' and the public's understanding of how their activities affect the Earth, as well as how geological events are controlled by larger-scale Earth dynamics. The overview of Earth science events also provides users with a holistic perspective to think of the Earth.

By creating Google Earth Geoscience Video Library, the authors developed a way to organize videos showing all kinds of Earth system processes, events, and objects, which are carried via the FB and non-FB videos. By projecting these events at different scales, giving different ways of seeing these events, and scaffolding users' understanding of them, we sought to give learners a more intuitive field-based experience of geoscience and how the Earth works. From the survey results, we see that students mostly liked the design of GEGVL and its function, as well as which features they did not like about it. They liked the 3D geographic context, enjoyed exploring it, appreciated the abundance of interesting information, liked being able to control what and when to use it, and appreciated its easy access and eye-catching qualities. The advantages of how GEGVL organizes videos were captured in comments such as "GEGVL enables me to learn a lot more without having to search in a different web browser" and "being able to view videos based on its geographic location is awesome". On the other hand, the results also showed that a small minority of students did not benefit from GEGVL, or at least they were not as excited about the experience as the majority. Some students reported that they are more likely to use the GEGVL "whenever it is useful for something they are doing". The greatest drawback of the GEGVL seems to be that students do not want to download the Google Earth app. Some students suggested that we can add a list of all GEGVL content in a separate browsable format.

Understanding Earth science concepts, especially the Earth systems, can be challenging [8–10]. The spatial nature of Google Earth-like interactive virtual platforms provides a unique opportunity for geoscientists and educators to enhance student understanding of Earth science concepts and how Earth systems work. Many geoscientists have already taken advantage of Google Earth to create field-based educational materials such as climate-related data visualization [48,49], and virtual geological mapping materials with Google Earth [23]. The major advantage of a Google-Earth-like platform from our assessment and in our experience is the freedom of exploring global scale 3D terrains in detail, which visually creates a virtual experience of exploring the Earth and allows students to observe the physical world that they normally cannot see. Additionally, the interactive nature of the platform helps them visualize interconnections, changing scales, multiple components, and feedback of different components of Earth systems. This provides a powerful geoscience learning environment for students and many teachers have already taken the advantage of this environment, but the key question is how to maximize the benefits. GEGVL provides one solution for teachers to enhance the benefits of videos in this environment to engage students and help students foster a more holistic view of the Earth. However, how to best take advantage of the platform in teaching Earth science is still unclear, and future research is still needed.

Orion [5] argued that the Earth systems approach is optimal for teaching Earth science but that the implementation of this worldwide is very limited. The design logic of GEGVL may provide a new way of incorporating Earth systems education for geoscience classrooms, allowing teachers to show pertinent events and objects to students. Teachers can also use the opportunity to explain the relationship of an event to the Earth system. In order to increase the usefulness of GEGVL to their geoscience courses, teachers can build on the GEGVL file to create their own geoscience video library demonstrations as well. Using the Google Earth pro tools, one can also add new icons and videos or add new content to the existing icons in GEGVL. We set up a page to share basic steps of how to add videos to Google Earth: https://utdgss2016.wixsite.com/utdgss/gegvl, accessed on 11 May 2022. The code needed and later updates of GEGVL can be found at https://github.com/Ning-utdgss/GEGVL-Updated.git, accessed on 11 May 2022. We continue to add GeoEd videos to GEGVL and make the GEGVL freely available so that geoscience educators and others can revise it for their own classes and purposes. We see this as a novel and trustable way for geoscientists to disseminate GeoEd videos they made. However, GEGVL as presently implemented on a Google Earth pro desktop version has many limitations. One of the greatest problems is that the Google Earth platform crashes fre-

quently when running the GEGVL Google Earth desktop platform when a large number of videos are embedded in it. This causes memory issues, leading to the platform crashing. The online version of Google Earth is powerful but does not provide the same functions as the desktop version [25], which are important for the design of GEGVL. Thus, creating a web-based platform for GEGVL is a more promising way to solve the "frequent crashing" problem. In this sense, the GEGVL is served as a protocol for the later platform to receive feedback from the students, educators, researchers, and geoscience community to improve the design for the later platform.

Another challenge concerns finding potential videos for inclusion in GEGVL and reviewing these for scientific quality. At present, only the UTD team proposes possible videos to include, and only the second author reviews them for scientific quality. GEGVL would benefit tremendously if communities of "GeoEd video finders" and geoscience experts took over these functions. A large, broad network with diverse expertise—from beginner to expert—is needed for the first task, and a smaller, more expert group is needed for the second task. We are a long way from establishing either community outside of UTD and welcome suggestions about how to address these two challenges. Video review will hopefully evolve so that it becomes more similar to the way that scientific manuscripts are reviewed today. Alternatively, something along the lines of how Wikipedia reviews new articles is needed. It may be that a new kind of video journal or citizen science-inspired project is needed to accomplish this task. Participation as reviewers in such a project may also provide a low entry threshold outreach opportunity for geoscientists and advanced geoscience students.

In addition, there is a major limitation in terms of the experiment design. Considering the population of college students, the surveyed sample size is not large enough to be representative. The reason is partly because of the issues related to the Google Earth platform that we mentioned earlier. This limits our ability to perform formal assessments and collect quantitative data in a large group. In addition, the survey is designed as a preliminary assessment to understand how college students think of GEGVL and receive feedback from them to improve the design of the GEGVL. Thus, we focused on mostly the sample sources than on the sample size. We limited our selection bias, randomly recruited our participants, and controlled our sample source by confirming their identity and asking demographic questions. Considering the diversity of gender, majors, and habits of using geoscience videos, we believe the survey sample is representative enough for our preliminary data.

Additionally, there are other field-based digital platforms aiming at the similar goals of GEGVL, such as iVFT made by the Penn State University team [30], providing mixed reality experiences including some beyond human experience about Earth, and Virtual Field Trips (VFTs) made by an ASU team [47], creating an immersive experience with interactive functions and 360° camera which also embeds many field-based videos. GEGVL is a less structured platform, compared with other existing VFT platforms. GEGVL gives more flexibility, information, and control of the content to teachers. Most VFTs have a fixed region, and once created, the contents are difficult to update. It is also hard for VFTs to reach global audiences. Unlike the works of others, GEGVL is not designed for direct use in the classroom but for teachers or students to search, find, and explore videos they like or think useful so that they can create their own "mini-GEGVL". By removing or hiding irrelevant videos and creating their own Google Earth kmz file, teachers can show students a customized set of videos in GEGVL for their classes. Since GEGVL is designed to organize videos, it can easily be updated to include new videos made by other geoscientists, making it easier to create a community for reviewing and recommending videos. By cooperating with organizations such as SERC, IRIS, or NASA, GEGVL can easily scale up to include more geoscience topics.

As GEGVL is designed as an unstructured and flexible video organizing platform, its educational effectiveness largely depends on how teachers use it. As Lombardi et al. [13] summarized in their recent review, scaffolding is critical for robust learning outcomes in active learning. Therefore, it is important for teachers to remember to be selective when using GEGVL for their classes and guide students while using GEGVL. Teachers may not want to direct their students to use GEGVL but create their own "mini-GEGVL" files which contain only relevant videos to support their lectures or assignments. This is also true for the field-based in-person experience. Although there are limitations of in-person field education, it provides a unique experience and benefits that a virtual environment cannot provide [1,15,41]. The most authentic sense of place and emotional experience is gained via in-person experience in the field [50]; even the most immersive virtual field trip cannot provide the same level of emotional experience and cognitive outcome [51]. Thus, GEGVL is only a supplement for in-person field trips, but the development of GEGVL can probably help geoscience educators efficiently find pertinent videos to prepare students for fieldwork or field trips. In addition, other educational methods such as Geoscience Concept Maps can be added to help explain connections among the videos (e.g., Vasconcelos et al. [52,53]). For example, selected videos in GEGVL may help explain a particular geoscience concept map, by using combined FB and non-FB videos as examples and scaffolding their learning, or by using GEGVL and selected videos and the virtual Earth environment to show events revealing how the feedback loops work to support the understanding of complex system behaviors in STELLA (now called isee systems, a complex system simulation and analysis software) [54]. Overall, the flexibility and scalable nature of GEGVL make it easily work with other educational designs.

Geoscientists play key roles in providing information and knowledge about local and global environmental risks and natural hazards. However, there are not enough geoscientists worldwide who interact well with the public, because many do not have the motivation or skills for such non-academic communication [5]. GEGVL provides a flexible way for scientists to join such efforts. The role of the geoscientists in contributing to GEGVL development includes recommending videos, reviewing comments, and making scientifically accurate videos about their research and local geology. Significant efforts should, therefore, be invested by geoscience researchers and professors to undertake a deep change at all levels of the university [5,23,52]. Applications such as GEGVL can help this transaction. More specifically, the GEGVL model can be useful for (1) attitudinal change research, (2) changing expertise; (3) geoethics, (4) interacting with the public, (5) enhancing the involvement of geoscientists in geoscience communication and education, and (6) supporting geoscience curriculum at all levels.

## 6. Conclusions

There is an urgent need for increasing Earth science literacy and improving student awareness of Earth system science and geoscience [2,5,55]. Individuals and groups create a wide range of geoscience videos involving different geoscience disciplines with different time and spatial scales that are aimed at different levels of learners and for different purposes. Many of these videos are about specific events, such as volcanic eruptions in Hawaii, or places such as Yellowstone National Park, that people are familiar with and want to know more about. Many event-based and object-based videos are similar to being on a field trip, albeit much shorter. Videos about events and objects can be projected to specific locations on Earth. By projecting these field-based videos in the virtual Earth environment and linking them with other geoscience educational videos (non-field-based videos), we can better explain the geoscientific concepts behind these events and objects. The design framework of Google Earth Geoscience Video Library (GEGVL) and students' feedback about the GEGVL are useful for creating more effective Earth science educational tools. The design of GEGVL provides a new way to build an Earth science educational tool for the 21st Century.

**Author Contributions:** Conceptualization, N.W.; methodology, N.W. and M.L.U.; formal analysis, N.W.; investigation, N.W.; resources, N.W., K.M.S. and R.J.S.; writing—original draft preparation, N.W.; writing—review and editing, N.W., R.J.S. and M.L.U.; visualization, N.W.; supervision, R.J.S. and M.L.U.; project administration, N.W.; funding acquisition, R.J.S. All authors have read and agreed to the published version of the manuscript.

**Funding:** This research was funded by the National Science Foundation, grant NSF-DUE1712495 to R.S.

**Institutional Review Board Statement:** Data collection from students at the University of Texas at Dallas was conducted following UTD Institutional Review Board (IRB) Protocol IRB-21-70 (approved 10 November 2020).

**Informed Consent Statement:** Student participants provided informed consent, per the UTD IRB protocol above.

**Data Availability Statement:** The following supporting information about the code for placemarks and updated GEGVL files can be downloaded at https://github.com/Ning-utdgss/GEGVL-Updated.git and https://utdgss2016.wixsite.com/utdgss/gegvl.

**Acknowledgments:** We would like to thank the editors and the reviewers of *Geosciences* for this paper. Thanks to assistance from Somtochukwu Nnachetta, Geoscience Studios colleagues, and other professors, organizations, and groups who have contributed to the creation of geoscience educational videos. This is UTD Geoscience contribution # 1686.

**Conflicts of Interest:** The authors declare no conflict of interest.

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
