# Peer review of "Google Earth Geoscience Video Library (GEGVL): Organizing Geoscience Videos in a Google Earth Environment to Support Fieldwork Teaching Methodology in Earth Science"

_geosciences, doi:10.3390/geosciences12060250_

Round 1

Reviewer 1 Report

This is a clear and well documented discussion of a new platform. While it may not have the biggest audience, the paper does what it sets out to do. A few minor points as listed below.

27 - I absolutely accept that improving student understanding is important, but I don’t know if the evidence is there for the merits of improving public understanding. 

31 - As with the previous, I am unsure that improving public understanding leads to increased sustainability

60 - videos, not video

85 - there is a big difference between K and college - what audience are you aiming for?

94 - ‘their YouTube or Google search for’?

109 -‘to no one region’?

130 - GeoEd or geode?
148 - do you have evidence it is fun?

202 - missing ‘to’

265 - it would be good to reflect on how representative these students are of your target audience

357 - probably ‘implemented’

396 - see points about line 27 above

Author Response

To Reviewer 1

Thank you so much for your encouraging words and appreciate your detailed review feedback! The following are my response to each item:

27 - I absolutely accept that improving student understanding is important, but I don’t know if the evidence is there for the merits of improving public understanding. 

RESPONSE: Interesting point. Thank you. I guess one of the most mentioned aspect is that how public literacy of the Earth science, especially the Earth system science literacy is important for solving the climate change, sustainability and environmental issues. (E.g. Wysession et al., 2012, J of Geo Ed). However, it is just a hypothesis, indeed I do not see any strong quantitative evidence showing how the increase of public understanding of ESS has solved the problem, some local examples talking about environmental knowledge led positive actions but it is not focused on ESS. Especially this paper focuses on the education not the public understanding of science, therefore, we have removed the word ‘public’.

31 - As with the previous, I am unsure that improving public understanding leads to increased sustainability

RESPONSE: Thanks, again, for the claim itself, I think there is some room to argue. But, what I know was mostly indirect, small scale and hypothetical evidence too (e.g. Dovers and Hussey, 2013 Environment and sustainability: a policy handbook. Reid, et al. 2010, on Science). Thus, considering this paper focuses on the education not the public understanding of science, therefore, we have removed the word ‘public’.

60 - videos, not video

RESPONSE: Yes, thank you!

85 - there is a big difference between K and college - what audience are you aiming for?

RESPONSE: For the assessment of this paper, we focus on the college students mostly. However, for the geoscience video library, it is a platform can serve a large range of students, depending on what videos put into the platform. We actually have been tested in both 8th grade teachers and college level students, we got good feedback from both. However, we only reported the evidence of college students in this paper, since we only filed IRB human subject research for college students.  Therefore, for the GEGVL, as a digital infrastructure for organizing videos, itself does not have limitations regarding teaching levels. As long as teachers want to use the proposed strategy and videos, they can insert videos by themselves to custom the GEGVL for their curricula.

94 - ‘their YouTube or Google search for’?

RESPONSE: Thank you. Yes, the sentence is incomplete. It should have been ‘their YouTube or Google search for geoscience topics…’

109 -‘to no one region’?

RESPONSE: Revised the sentence to ‘not pertaining to any particular region’

130 - GeoEd or geode?

RESPONSE: Thanks! It is GeoEd.

148 - do you have evidence it is fun?

RESPONSE: Yes, the evidence is from qualitative data in the results section. See ‘figure 6’ the second and seventh frequently mentioned comments (fun, excited) about the experience of using GEGVL.

202 - missing ‘to’

RESPONSE: Yes, thanks. Added.

265 - it would be good to reflect on how representative these students are of your target audience

RESPONSE: Great point. We add the discussion at Line: 410 to 420.

357 - probably ‘implemented’

RESPONSE: Yes. Thank you! Fixed.

396 - see points about line 27 above

 RESPONSE: Yes, replaced the public with student

Reviewer 2 Report

Overall

The tool described in the manuscript is a valuable resource on its own and offers an opportunity to address some of the broader conceptual and logistical challenges of providing PBE resources, particularly in the form of virtual representations of places.  I would encourage the authors to develop a bit the formal analysis of challenges and solutions. As note the tool is locally supported by a small research group which comes with significant challenges to support, adoption and sustainability.  The following comments are offered to help guide a revision to give the paper greater scope and perhaps serve as a nucleation point for building a community around geoscience video and virtual resources.

There are important science of learning questions about virtual PBE for which there is little data that would allow evidence informed guidance.  The primary one relevant here is: in what ways can virtual experiences replace in-person experiences (and are there ways in which virtual experiences only weakly support learning or are even counter-productive)? The authors note the benefits of their tool, but not potential down sides.  An important question to address is, are there ways to design the virtual experiences to mitigate threats to learning and enhance the opportunities that virtual experiences provide?

A spatially organized collection of videos offers a very different one from virtual field trips (e.g. https://vft.asu.edu – note to authors there are a fair number of videos embedded here) where videos may be embedded along organized collections of other information about a place. Well designed virtual trips are quite labor intensive. An important question is, will unstructured presentation of videos support learning? And does it support it in the same ways that PBE does?

The manuscript implies in a number of places that this tool can help with important learning challenges.  I don’t disagree, but feel as written the manuscript underestimates the challenges: 

1) Unguided grazing of content tends not to result in robust learning outcomes. One of the reasons research on the value of active learning is mixed is that guidance on how to design active learning is not completely established, but from existing research it is clear that structured guidance is critical. I would encourage the authors to review the summary of research on active learning and field-based education that can be found in Lombardi et al 2021 The curious construct of active learning. Psychological Science in the Public Interest, 22(1), 8-43.

2) Understanding systems is quite challenging, and agree that this resource could support learning.  That said simply providing videos is unlikely to enhance understanding of Earth systems.  There is the opportunity to use the spatial nature of Google Earth to highlight aspects of systems and support learning challenges (interconnections, scale, multiple components, feedback). The authors may wish to add their thoughts on how to do that, or barring that, at least raise it as an important problem to be addressed by the community.  

The authors note the tool is compatible with concept maps. This point could be broadened to consider how to integrate their tool with other spatial approaches to supporting learning.

A final point.  As the authors note, there are challenges to building a tool on Google Earth. My understanding is that Google has history of changing underlying code of its products with little warning as well as dropping support for its tools. That seems a weak foundation to build a potentially important education resource.

Minor edits and line edits:

The references are somewhat limited, I would encourage the authors to review both volumes of Earth and Mind edited by Manduca, Mogk, and Kastens, as well as work by Eric Riggs and Alexander Klippel.

Abstract” show that the design logic of GEGVL is promising virtual PBE organizer for “ missing “a”

Peixoto, E.; Pedro, L.; Vieira, R. Transmedia in Geosciences Education. Geosciences 2022, 12, 171: this study does not support the claim “the important roles that active learning and outdoor education play in these efforts because these methods especially engage students and enhance learning outcomes”

“could better manage cognitive load via multimedia designs [22]” this is a stretch. Its not clear that being in person creates greater cognitive load than viewing videos of that place.  Suggest rephrasing in line with Mayer’s work that it is important to design videos in such a way that they reduce cognitive load.

“video search time and trustworthiness of content [26,27]” the paragraph context implies the data comes from geoEd videos but appears to be a more general observation; suggest making that clear.

“We have done this on an ad hoc basis, but it is clear that more thought needs to be given to how best to peer review geoscience videos before putting these into GEGVL or any other educational video library, so that teachers and students can be confident in video content.”  SERC has extensive experience reviewing educational materials, and might be a model for both building the necessary community as well as the standards for geoscience videos for learning.  The authors might consider the value of decoupling the search/view and review processes, to have a reviewed repository (with content that does not change or disappear as could happen with links to external video sources) with tags that can be imported into their system.

“We highlight PB videos because this is how people learn things naturally [1,11,31]”  as written this claim is unfounded.  Also, although clearly intended as a good thing, it is not clear what claim is being made to call the learning “natural”.

Suggest doing a search to find where autocorrect inserted “geode “ in for GeoEd.

“Allow teachers and learners to roam the Earth to freely choose what to learn. This approach can also help diverse groups with different backgrounds, interests and geoscientific knowledge.” As noted above this claim is not well supported by the reseaerch and implies simply providing the resource is “active learning” and therefore a sufficient solution to significant learning challenges.

“Using multiple representatives has been proven as a very effective teaching method [34].“ Review this claim.  Multiple videos is not what Ainsworth meant by multiple representations.

“A single icon location for is not appropriate in this situation,” missing or extra word?

Figure 4. Demographic and background survey of the participants (N=71).”  Graphical presentation of the data is redundant with text and does not add significantly to understanding.

“Seventy students like the GEGVL” should be past tense.

The data format does not appear to match the survey ( viewed on May 23).  The version I viewed had endorse with yes or no to the value of GEGVL (“Do you think you will…) no option for strongly agree etc.

“Exploring videos on GEGVL stimulates this activity and provides a more intuitive way to increase their beyond-human scale experience about Earth systems and foster a better mental model for incorporating natural phenomenon and integrating more information about Earth system science.” This should be phrased as a hypothesis (a not unreasonable one); there is not an extensive research basis to support this claim.

“Orion [38] argued that the Earth systems approach is optimal for teaching Earth science but that the implementation of this worldwide is very limited. The design logic of GEGVL can provide a new way of incorporating Earth systems education for geoscience classrooms because it allows teachers to show pertinent events and objects to students and explain the relationship of an event to the Earth system. “ As noted above systems are challenging to understand.  This wording suggests that learning about systems can be addressed easily.

“enhancing cooperation of geoscientists in geoscience communication and education” suspect that “cooperation” is not what you meant here, but not sure what you were trying to convey to offer an alternative.

“In addition, ideally, other educational methods such as Geoscience Concept Map can be embedded into this model to explain connections among the videos (e.g. Vasconcelos et al.[39]).” This sentence does not seem to fit.  Perhaps this could be moved to discussions of supporting systems learning and you could explain how you imagine, within the constraints of your tool, you might implement construction of concept maps by students. It also raises the broader questions of what other sorts of education resources or learning tools can be integrated (and which ones can not).

Author Response

Overall

The tool described in the manuscript is a valuable resource on its own and offers an opportunity to address some of the broader conceptual and logistical challenges of providing PBE resources, particularly in the form of virtual representations of places.  I would encourage the authors to develop a bit the formal analysis of challenges and solutions. As note the tool is locally supported by a small research group which comes with significant challenges to support, adoption and sustainability.  The following comments are offered to help guide a revision to give the paper greater scope and perhaps serve as a nucleation point for building a community around geoscience video and virtual resources.

RESPONSE: Yes, thank you so much for the encouraging words. We will build on this idea and create a server-based geoscience video library platform so that we can do bigger scale and formal assessment.

There is important science of learning questions about virtual PBE for which there is little data that would allow evidence informed guidance.  The primary one relevant here is: in what ways can virtual experiences replace in-person experiences (and are there ways in which virtual experiences only weakly support learning or are even counter-productive)? The authors note the benefits of their tool, but not potential down sides.  An important question to address is, are there ways to design the virtual experiences to mitigate threats to learning and enhance the opportunities that virtual experiences provide?

RESPONSE: Very insightful comment, thanks. Very good point, we should not have used the word, ‘provide an alternative PBE experience’ in the abstract ‘line 11’. We now use ‘supplementary PBE experience’. Just as the reviewer mentioned, the experience of field-based learning so far provides a unique experience for the learners, the GEGVL cannot provide the same experience as the in-person field experience. From our assessment, we see it triggers the motivation of people go to the field and see by themselves not replace it. Although as we mentioned in the paper, in-person field trip has some requirements, the benefits of field trips are also not replaceable by virtual field trips. We also added discussion about this: Please see line 448 to 462.

A spatially organized collection of videos offers a very different one from virtual field trips (e.g. https://vft.asu.edu – note to authors there are a fair number of videos embedded here) where videos may be embedded along organized collections of other information about a place. Well designed virtual trips are quite labor intensive. An important question is, will unstructured presentation of videos support learning? And does it support it in the same ways that PBE does?

RESPONSE: Thanks! We added a section to discuss this: Please see line 440 to 448.

The manuscript implies in a number of places that this tool can help with important learning challenges.  I don’t disagree, but feel as written the manuscript underestimates the challenges: 

1) Unguided grazing of content tends not to result in robust learning outcomes. One of the reasons research on the value of active learning is mixed is that guidance on how to design active learning is not completely established, but from existing research it is clear that structured guidance is critical. I would encourage the authors to review the summary of research on active learning and field-based education that can be found in Lombardi et al 2021 The curious construct of active learning. Psychological Science in the Public Interest, 22(1), 8-43.

RESPONSE: Thank you so much for sharing us with the article and insights. We should have made our point clearer, and we’ve added more into the discussion. We make it clear that using GEGVL in formal education should be dealt with like supplementary material for each class, or directly embedded into the classes. GEGVL expedites learning from the videos, and guidance should be added in formal education. Scaffolding by teachers, either in person or giving instructions about the learning purposes is important for learning outcomes. Also, GEGVL can be used as informal supplementary resources for extended learning, if students want. GEGVL placemarks and videos can serve as a starting point of learning more, such as leading the users to other peer reviewed papers or book chapters, etc. However, it is a great idea to add more discussion about in-person and virtual field trips. Please see line: 448 to 462.

2) Understanding systems is quite challenging, and agree that this resource could support learning.  That said simply providing videos is unlikely to enhance understanding of Earth systems.  There is the opportunity to use the spatial nature of Google Earth to highlight aspects of systems and support learning challenges (interconnections, scale, multiple components, feedback). The authors may wish to add their thoughts on how to do that, or barring that, at least raise it as an important problem to be addressed by the community.  

RESPONSE: Thank you, yes, we added some discussion about the Google Earth like spatial tools potential to help understand Earth Systems and their limitations. Please see line 354 – 371.

The authors note the tool is compatible with concept maps. This point could be broadened to consider how to integrate their tool with other spatial approaches to supporting learning.

RESPONSE: Thank you, yes, we add some discussion about how GEGVL can be integrated with other teaching methods to help understand Earth Systems and geoscience courses. Please see line 456 – 463.

A final point.  As the authors note, there are challenges to building a tool on Google Earth. My understanding is that Google has history of changing underlying code of its products with little warning as well as dropping support for its tools. That seems a weak foundation to build a potentially important education resource.

RESPONSE: Thanks, we agree. We now, besides stating this weakness, also highlight the design logic of this tool, using ‘Virtual Earth Environment to Organize Videos to Support Learning’, and we will move away from Google Earth and create a web-based platform with the same strategy. This is also a paper trying to get attention more potential helpers from the geoscience and education community to build a better future tool together. We also add discussion here: Line 421 to 439.

Minor edits and line edits:

#8 The references are somewhat limited, I would encourage the authors to review both volumes of Earth and Mind edited by Manduca, Mogk, and Kastens, as well as work by Eric Riggs and Alexander Klippel.

RESPONSE: Thank you, yes, we add more references from these works. Please see line: 30 – 50, and some are added in the discussion sections.

Abstract” show that the design logic of GEGVL is promising virtual PBE organizer for “ missing “a”

RESPONSE: Yes, we added some discussion about how GEGVL can be integrated with other teaching method to help understand Earth Systems and geoscience courses.

Peixoto, E.; Pedro, L.; Vieira, R. Transmedia in Geosciences Education. Geosciences 2022, 12, 171: this study does not support the claim “the important roles that active learning and outdoor education play in these efforts because these methods especially engage students and enhance learning outcomes”

RESPONSE: This is arguable. Although there is no single sentence directly mentioning the statement, Peixoto (2021)’s paper shows how transmedia activities enable students to actively seek information and critically think and solve problems (active learning) and learn by social context and field trip experience (outdoor learning). The result of the paper shows an example of how the combo can be effective in terms of enhancing learning and engaging students. 

“could better manage cognitive load via multimedia designs [22]” this is a stretch. It’s not clear that being in person creates greater cognitive load than viewing videos of that place.  Suggest rephrasing in line with Mayer’s work that it is important to design videos in such a way that they reduce cognitive load.

RESPONSE: Good point. We rephrased the sentence to ‘at the same time could use multimedia designs to manage cognitive load’.

“video search time and trustworthiness of content [26,27]” the paragraph context implies the data comes from GeoEd videos but appears to be a more general observation; suggest making that clear.

RESPONSE: Thank you so much for pointing this out! We have rephrased the sentence to “Current research shows that the two most mentioned concerns of K-12 teachers are video search time and trustworthiness of content [26,27].

“We have done this on an ad hoc basis, but it is clear that more thought needs to be given to how best to peer review geoscience videos before putting these into GEGVL or any other educational video library, so that teachers and students can be confident in video content.”  SERC has extensive experience reviewing educational materials, and might be a model for both building the necessary community as well as the standards for geoscience videos for learning.  The authors might consider the value of decoupling the search/view and review processes, to have a reviewed repository (with content that does not change or disappear as could happen with links to external video sources) with tags that can be imported into their system.

RESPONSE: Yes, good point! SERC does have those community reviewed activities and that is a great idea to add these reviewed videos and materials into the GEGVL. SERC also helped host several workshops about geoscience video making. However, based on our search and communication with SERC people, we haven’t seen that they have a community or standards for educational geoscience videos yet, although they have some valuable recommendations. It is a great idea to create a reviewed repository working with SERC, we will definitely consider it.

“We highlight PB videos because this is how people learn things naturally [1,11,31]” as written this claim is unfounded.  Also, although clearly intended as a good thing, it is not clear what claim is being made to call the learning “natural”.

RESPONSE: Good point. Thanks for pointing out the confusing part of this sentence. The citations are indicating how people learn better under context, so we rephrased the sentence as “We highlight PB videos because people can learn things better under real social and physical contexts [1,11,31].”. The three references provide the evidence of people can learn things better under real social and physical contexts.

Suggest doing a search to find where autocorrect inserted “geode” in for GeoEd.

RESPONSE: Yes, thanks, we have done the search and found one misspelling, and fixed.

“Allow teachers and learners to roam the Earth to freely choose what to learn. This approach can also help diverse groups with different backgrounds, interests and geoscientific knowledge.” As noted above this claim is not well supported by the research and implies simply providing the resource is “active learning” and therefore a sufficient solution to significant learning challenges.

“Using multiple representatives has been proven as a very effective teaching method [34].“ Review this claim.  Multiple videos is not what Ainsworth meant by multiple representations.

RESPONSE: Thanks for pointing it out! We are sorry that the sentence didn’t express what we mean about the multiple representations.

First, we do not intend to say that multiple videos are multiple representations. What we tried to indicate was that different creators may make different videos about one topic, which forms multiple representations of the same topic. Videos made by different geoscientists explaining the same topic can be very different regarding the considered scales, geological time span, angles, depth of the knowledge, and details of the topic, etc.. For PB videos on the same topic, the placemark in GEGVL can organize these different videos about the same location or event together (inside one placemark) to give students different representatives of the same topic or feature. For non-PB videos, different real examples (PB videos) become the different representatives of the same concept (non-PB videos). Plus, the virtual 3D environment itself commonly provides a unique dimension of the representative of an Earth feature.  Although the Ainsworth paper mostly focuses on multiple sensory presentations, like multimedia, she also mentioned how multiple representations go beyond that and can be many dimensions and different categories (page 2 and 3 of the paper).

Second, this may be arguable but based on our understanding, the claim ‘Using multiple representatives has been proven as a very effective teaching method’ has a citation directly from the Ainsworth article at page 3 ‘There is abundant evidence showing the advantages that external representations play in supporting learning. Much research has shown that matching the type of representation to the learning demands of the situation can significantly improve performance and understanding.’ and evidence has been listed in the rest of the page 3, page 4 and especially page 5 ‘a number of influential educational theories discuss the importance of MERs. For example, Dienes (1973) argues that perceptual variability (the same concepts represented in varying ways) provides learners with the opportunity to build abstractions about mathematical concepts. In cognitive flexibility theory, the ability to construct and switch between multiple perspectives of a domain is fundamental to successful learning (Spiro & Jehng, 1990). Research on analogical reasoning shows how comparison processes help people make new inferences (Gentner & Markman, 1997).’ Although there is no consistent evidence showing giving multiple representations always enhance learning, the DeFT model did support giving multiple representations in interactive platform and providing options for learners.

Therefore, we argue that this is an appropriate claim and the citation supports the claim.

Figure 4. Demographic and background survey of the participants (N=71).”  Graphical presentation of the data is redundant with text and does not add significantly to understanding.

RESPONSE: Removed the graph.

“Seventy students like the GEGVL” should be past tense.

RESPONSE: Removed the graph.

The data format does not appear to match the survey ( viewed on May 23).  The version I viewed had endorse with yes or no to the value of GEGVL (“Do you think you will…) no option for strongly agree etc.

RESPONSE: Oh yes, thank you so much for reminding us for this! Our survey link should be https://redcap.link/GEGVL-S2, the old link was a beta version for early internal discussion. Thank you so much for reminding us! The survey link has been fixed now.

“Exploring videos on GEGVL stimulates this activity and provides a more intuitive way to increase their beyond-human scale experience about Earth systems and foster a better mental model for incorporating natural phenomenon and integrating more information about Earth system science.” This should be phrased as a hypothesis (a not unreasonable one); there is not an extensive research basis to support this claim.

RESPONSE: Thanks a lot for the advice! Yes, we rephrased the term into “Exploring videos on GEGVL has the potential to stimulate this activity and provide a more intuitive way to increase their beyond-human scale experience about Earth systems and foster a better mental model for incorporating natural phenomenon and integrate more information about Earth system science.”

“Orion [38] argued that the Earth systems approach is optimal for teaching Earth science but that the implementation of this worldwide is very limited. The design logic of GEGVL can provide a new way of incorporating Earth systems education for geoscience classrooms because it allows teachers to show pertinent events and objects to students and explain the relationship of an event to the Earth system. “ As noted above systems are challenging to understand.  This wording suggests that learResponse about systems can be addressed easily.

RESPONSE: Thanks! We have rephrased the sentence into: Orion [38] argued that the Earth systems approach is optimal for teaching Earth science but that the implementation of this worldwide is very limited. The design logic of GEGVL provides a new way of incorporating Earth systems education for geoscience classrooms allowing teachers to show pertinent events and objects to students. Teachers can also use the opportunity to explain the relationship of an event to the Earth system.

“enhancing cooperation of geoscientists in geoscience communication and education” suspect that “cooperation” is not what you meant here, but not sure what you were trying to convey to offer an alternative.

RESPONSE: Yes, thanks. We replaced the word ‘cooperation’ with ‘the involvement’, so the sentence will be ‘encouraging the involvement of geoscientists in geoscience communication and education’

“In addition, ideally, other educational methods such as Geoscience Concept Map can be embedded into this model to explain connections among the videos (e.g. Vasconcelos et al.[39]).” This sentence does not seem to fit.  Perhaps this could be moved to discussions of supporting systems learning and you could explain how you imagine, within the constraints of your tool, you might implement construction of concept maps by students. It also raises the broader questions of what other sorts of education resources or learning tools can be integrated (and which ones can not).

RESPONSE: Good point! Thanks for the advice. We have moved the sentence to the section and add more discussion about the topic. Please see line: 456 – 463. Thank you so much for all your great advice and comments! Appreciate your help!